# Investigating the Impact of Electrolyte Flow Velocity on the Resistivity of Vanadium Redox Batteries: A Theoretical Analysis and Experimental Data Comparison

**DOI:** 10.3390/ma16134845

**Published:** 2023-07-05

**Authors:** Roman Kislov, Zekharya Danin, Moshe Averbukh

**Affiliations:** Department of Electrical and Electronics Engineering, Ariel University, Ariel 40700, Israel; romanki@ariel.ac.il (R.K.); zekharyad@ariel.ac.il (Z.D.)

**Keywords:** ion-membrane conductivity, math model, electrolyte velocity, battery internal resistance

## Abstract

The influence of electrolyte velocity over the ion-exchange membrane surface on ion and vanadium redox batteries’ conductivity was formalized and quantified. The increase in electrolyte velocity dramatically improves proton conductivity, resulting in improved battery efficiency. An analysis of conductivity was carried out using a math model considering diffusion and drift ion motion together with their mass transport. The model is represented by the system of partial differential together with algebraic equations describing the steady-state mode of dynamic behavior. The theoretical solution obtained was compared qualitatively with the experimental results that prove the correctness of the submitted math model describing the influence of the electrolyte flow on the resistance of the vanadium redox battery. The presented theoretical approach was employed to conduct a parametric analysis of flow batteries, aiming to estimate the impact of electrolyte velocity on the output characteristics of these batteries.

## 1. Introduction

In recent years, the demand for electrical energy storage has been on the rise [1,2]. In order to meet this demand, flow batteries have gained significant attention due to their unique advantages. These advantages include an extended service life, as there are no solid–liquid transformations involved, and the ability to store energy in separate tanks with different quantities of electrolyte solution.

One key benefit of flow batteries is the ability to independently design their capacitance and power. The power of a flow battery is determined by the size and number of active cells, allowing for scalability. On the other hand, the battery capacitance is determined by the total volume of electrolytes in the separate tanks. This flexibility enables the tailoring of flow batteries to meet specific energy storage requirements.

Furthermore, flow batteries provide enhanced control capabilities that are not achievable with conventional electrochemical devices [3]. This improved control allows for better management of energy flow and utilization, making flow batteries a promising solution for energy storage applications.

The flow battery can provide precise measurement for its electro-motive force (EMF), which indicates the exact state of charge (SoC) or state of discharge (SoD) [4]. By knowing the SoD, the optimal control of electrolyte flow can be achieved, resulting in improved battery efficiency [5].

The aforementioned advantages have led to a growing interest in the development and application of flow batteries in the energy industry. However, to fully realize their potential, flow batteries require detailed analysis and improvements in electrolyte flow regulation. The typical design of a vanadium battery is shown in Figure 1.

Redox batteries require the consideration of the consistent flow of electrolytes through the electrodes to accurately describe battery behavior [6,7]. Some research works have shown that reducing the flow velocity below a certain threshold results in a significant decrease in power due to an increase in the battery’s internal resistance [5]. The authors [5] assumed that the precise cause of this phenomenon is the creation of an inactive ion layer on the electrode surface that increases the internal resistance and battery power. However, a detailed theoretical analysis of the electrolyte velocity on the abovementioned effect was not performed. Other studies suggest that the resistance is related to the ability of the membrane to conduct ions. In their research, the authors of [8] conducted a study on the conductivity of four different types of ion-exchange membranes. They analyzed the high-frequency resistance in relation to the varying acid content of electrolytes. In a subsequent study [9], membrane conductivity was investigated as a function of vanadium and sulphite ion (${SO}_{4}^{2-}$) concentrations. A comparative study on the proton conductivity provided by cation and anion exchange membranes, as well as porous separators in sulphite acid solution, $H_{2}{SO}_{4},$ was conducted in [10]. The results of this investigation highlighted the influence of different material thicknesses on proton conductivity and demonstrated the superiority of cation exchange membranes over separators. The research conducted in [11] focused on the transport properties of the Nafion™ 212 membrane, examining its potential for increasing conductivity with the usage of twelve different alkali cations in flow batteries. Their research aimed to understand the membrane’s performance in facilitating ion transport. The investigation in [12] also contributed to the field by studying the influence of sulfonated fluorinated polyarylene (SFPAE) membrane thickness on VRB efficiency. Their findings revealed a non-monotonic relationship between VRB efficiency and membrane thickness. Specifically, they observed that a thickness change from 28 µm to 45 µm resulted in the lowest efficiency. However, further increasing the thickness to 80 µm improved the efficiency.

The study described in [13] incorporated the influence of electrolyte velocity on the description of the VRB functionality which was taken into consideration in the theoretical model. In addition to the theoretical description of the effect of electrolyte velocity, Refs. [6,14] suggested the optimization of the electrolyte flow control. The reason for the optimization is the need to reduce to a minimum the polarization effect on the membrane and electrode surfaces causing an increase in the battery resistance. The difference between the two optimization approaches is in the chosen criteria. A criterion intended to maximize the battery energy during discharge while minimizing the battery energy during charge was chosen in [13], while [6] used a criterion whose goal is the maximization of battery energy efficiency together with ensuring battery endurable operation under fast alternating load [14]. To achieve the goal of optimization, both abovementioned works took into consideration the hydraulic losses escorting the electrolyte current through the cell’s electrodes, which are made from porous media—carbon felt as a rule. These losses cannot be neglected, as shown in multiple experiments, and compose a significant part of the total energy waste in the flow battery.

An additional factor that affects battery resistance is the limited ability of the ion-exchange membrane to conduct protons. This is because the conductive capability of the membrane has a more significant impact on resistance than the rate of electrode reactions, for two reasons. First, the design of the carbon felt frames significantly enlarges the surface area of the electrode. Second, the rate of the electrochemical reactions on the electrode is significantly high. Despite this, the conductivity of proton exchange membranes remains relatively low, and the maximum achievable current density is still in the range of $40–60\;mA/{cm}^{2}$. Moreover, since the membrane is located across a cell board, which has a smaller area, while the electrode–felt filaments fill the entire volume of the cell, the influence of the membrane appears to be much more significant. The effect of membrane conductivity on battery resistance has been studied in numerous scientific works. For example, Ref. [15] provided a model of VRB resistance taking into consideration different ion transfer rates in the electrolyte bulk and through the membrane. The work described in [16] presented a model of a VRB-based Simulink (MATLAB) with the account of vanadium ions and protons through the cation-exchange membrane. The same investigations of the battery’s ability to provide output current vs. membrane conductivity parameters were conducted in [10,17,18,19], which provided experimental results regarding the influence of different separator material conducting properties, its thickness, and porosity on battery efficiency. It is worth emphasizing that [16,17] suggested describing the electrical potentials on the electrodes with the Nernst equation and the time-dependent characteristics of ion concentrations in the electrolyte bulk.

However, the local charge transport through the direction of electrolyte flow over the membrane surface is not discussed in detail. The theoretical investigation of VRBs was based on semi-empirical models [13,16,17,20] of the membrane’s resistivity, which was determined empirically without a strict theoretical foundation. However, this approach cannot rigorously investigate the influence of flow speed on battery resistivity. While the model proposed by [20] attempts to provide a more accurate theoretical explanation of battery resistivity, it is only applicable to porous media electrodes and is based on equations with empirical coefficients obtained from experimental tests, limiting its applicability to different types of batteries. Furthermore, the utilization of a 1-D approximation in the model fails to adequately account for the conservation law of electrolyte matter. Consequently, the model’s ability to accurately depict the influence of electrolyte flow on battery resistance is significantly compromised.

Therefore, the aim of this work is to develop a new theoretical model that describes the resistivity of the membrane as a function of the electrolyte velocity in the vicinity and ion current flowing perpendicular to the membrane surface under the diffusion and drift forces. To achieve this, a special analytical approach was created based on a system of partial differential equations (PDEs) describing the electrical properties of the cell. The boundary conditions for the solution of these PDEs include the concentration of protons on the electrodes and membrane surface, as well as the requirement for continuity of ion flow through the membrane.

## 2. The Modeling of the VR Cell

The main approach employed in this research work is the mathematical modeling of VR (vanadium redox) cells, which aims to describe the relationship between the cell (battery) resistance and electrolyte flow across the membrane surface. The model is constructed by considering the fundamental flow of ions within the cell, evaluating the electrical field across the electrode surfaces and the membrane, and estimating the density of charges and intensity of ion flow.

### 2.1. Principal Scheme of a Flow Battery Cell and Problem Definition

The functionality of VRBs (Figure 1) can be explained in detail by the principal scheme of one individual cell in the battery. The construction of a typical cell includes two compartments, each with fibrous electrodes made from carbon filaments or carbon nanotubes [21,22,23]. These compartments are separated by a proton-exchange membrane, typically Nafion-type [24]. Two different electrolyte solutions, one containing *V*^+2^ and *V*^+3^ ions and the other including *V*^+4^ and *V*^+5^ ions, are pumped through the electrodes of each compartment, causing corresponding electrochemical reactions that generate the battery’s electromotive force and output cell voltage. The detailed electrochemical reactions for the charge–discharge processes in each compartment are described below:(1)$$\begin{array}{l} \mathrm{Positive}\;\mathrm{electrode}:\;{VO}_{2}^{+}+e^{-}+2H^{+}\;↔\;{VO}^{2+}+H_{2}O,\;E^{0}=+1.004V\;\\ \mathrm{Negative}\;\mathrm{electrode}:\;V^{2+}↔\;V^{3+}+e^{-}\;,\;E^{0}=-0.255V\\ \mathrm{Overall}\;\mathrm{reaction}:\;{VO}_{2}^{+}+V^{2+}+2H^{+}\;↔\;{VO}^{2+}+V^{3+}+H_{2}O,\;\;E^{0}=+1.259V \end{array}$$

The primary function of a proton-exchange membrane inside a cell is to separate the electrolytes in the positive and negative compartments while allowing for the free flow of protons (H^+^) between them. As indicated by the equations for the continuation of the electrochemical process and the generation of constant electric current from/to a cell, H^+^ ions are produced on one electrode while they are absorbed on another. Consequently, proton transportation through the membrane is an essential feature that determines the cell’s ability to provide electrical power and its efficiency.

A sectional drawing of a battery cell is shown in Figure 2. The electrodes are represented for simplicity as flat plates 1 and 2 located on both internal boards of the cell, whereas the ion exchange membrane is in the middle, between the electrodes, dividing the cell into two compartments. Both spaces of the compartments are filled with positive and negative electrolytes which participate in electrochemical reactions (1).

In Figure 1, several key components are indicated:X, Y, Z: geometric coordinates used to represent the spatial dimensions.Membrane m: the membrane with a thickness *δ_m_* and a diffusion coefficient of ion conductivity, *D_m_*.Diffusion coefficients: *D*_1_ and *D*_2_ represent the diffusion coefficients of proton ions within the negative and positive compartments, respectively.*V*—the electrolyte velocity over the membrane surface in the X direction.*L*—the distance between the membrane and electrodes.*L_X_*—the length of the cell in the X direction.

In the diagram, the X and Y origins coincide with the left end of the membrane, while the Z origin is positioned at the lower left corner of the negative compartment.

The *X*-axis is directed along the plates, and the *Z*-axis is perpendicular to them. For simplicity, edge effects are ignored, and the plates are considered infinite, while a section of length *L_X_* along the *X*-axis is used in the calculations. The space of the compartments is filled with electrolytes and is divided into two equal parts with a width of l in the direction Z. The membrane has an effective thickness of *δ_m_* << l and allows protons to pass through it under the concentration difference only in the Z direction. The proton penetration through the membrane is characterized by the diffusion coefficient *D_m_*. The charge of the protons is denoted as *q*. The negative and positive electrolytes are pumped into their respective compartments and at X = 0 have the same ion concentration as in the tank. The tank has two reservoirs that accumulate both electrolytes after some discharge during the battery cell’s operation, leading to a decrease in the active ion concentration. Two electrolytes move with the same velocity v in the X direction. During the movement of ions over the membrane surface, the concentrations of ions are changed since a battery either provides current to the load or accept current from a charger. At X = 0, near both surfaces of the membrane, the difference in the concentrations is denoted as *δn_0_* and is determined by the concentration of electrolytes in both reservoirs of tank. On the membrane surface in the X-direction, the concentration of ions represents the boundary conditions required for the PDEs. These boundary conditions are obtained as the only allowable stable steady-state solution of the concentration, which is represented as the exponential distribution of ions on the membrane surface. The ion concentrations in another space of the cell under (*n*_1_) and above the membrane (*n*_2_) are the function of coordinates x and Z and will be obtained based on the PDE. This happens due to the electric field that arises from the separation of charges and ion flow *j* across the membrane.

The decision problem is formulated based on the following assumptions:In a vanadium redox flow battery, the individual cell voltage is relatively low, typically around 1.4 V. To achieve higher voltages, multiple vanadium cells are connected in series (N-serial connection). Consequently, the total battery voltage is determined by multiplying the voltage of a single cell, *V*_0_, by the total number of cells in the series string.On the other hand, the battery current is equal to the current of each individual cell. In other words, the current passing through the battery is the same as the current flowing through each cell in the series. The flow of current within each cell is facilitated by the movement of ions, specifically protons, through the proton-exchange membrane.The cell’s electrical behavior is primarily determined by the motion of protons, and therefore only protons are considered in the explanation.Protons are generated through the electrochemical transformation of vanadium ions on one electrode and are absorbed as an essential element required to continue the electrochemical reaction on another electrode.Protons move from one compartment to another under a concentration difference and electric field, which are mainly produced in the vicinity of electrodes and the membrane. The initial concentration is prescribed by the electrolyte velocity at the entrance of the cell.The membrane transportation property mainly defines the current density of the cell since its area is much lower than that of porous electrodes. In addition, the current density in the battery is influenced by the boundary conditions at the electrodes, which are defined by the potential difference between them.

The objective of this investigation is to determine the relationship between the resistance of the battery and the velocity of the electrolyte. We aim to accomplish this by creating a mathematical model that considers the spatial distribution of protons and their current density in the battery. To achieve this goal, we need to estimate the electric field produced near the electrodes and membrane.

### 2.2. The Density of Electric Charges and Proton Transportation

Let us analyze the spatial distributions of the proton flux, *j*(*x*,*z*), and concentration, *n*(*x*,*z*), near the membrane. We will assume that all quantities are uniform along the *Y*-axis, which simplifies the problem to two dimensions. Consider an electrolyte element with the shape of a parallelepiped, with sides *dx*, *dy*, and *dz*, as shown in Figure 3. The law of conservation of the proton flux can be expressed as follows:(2)$$\left(j(x,z+dz)-j(x,z)\right)Ydx=V\left(n(x+dx,z)-n(x,z)\right)Ydz$$

It is important to note that the left side of Equation (2) represents the difference between the incoming and outgoing streams of protons that are perpendicular to the (dx-dy) face of the electrolyte element. On the other hand, the right side of Equation (2) contains the difference in the proton flows in x-direction induced by the velocity V of the electrolyte.

From Equation (2), we can deduce that:(3)$$\frac{\partial j}{\partial z}=V\frac{\partial n}{\partial x}$$

As the flow of protons is continuous, the value of the proton flux, *j*, remains constant at z → l ±δm/2 (on the lower and upper surfaces of the membrane). This implies that the derivative, ∂j/∂z, changes its sign when the flux, *j*, crosses the membrane surface in the z-direction.

If we denote the proton concentration, *n*(*x*,*z*), as *n*_1_(*x*,*z*) for the coordinate range 0 < *z* < *l* and as *n*(*x*,*z*) *= n*_2_(*x*,*z*) for the range *l* < *z* < 2*l*, then we can approve the following:(4)$${\left(V\frac{\partial n_{1}}{\partial x}\right)}_{z\to l-\delta_{m}/2}={\left(-V\frac{\partial n_{2}}{\partial x}\right)}_{z\to l+\delta_{m}/2}$$

The membrane is typically considered to be infinitely thin, since $l>>\delta_{m}\;$ for practically usable cells. However, this assumption may not be held for the estimation of the proton concentration gradient inside a membrane.

Equation (4) indicates that the sum of the two concentrations on the lower and upper membrane surfaces in both compartments does not depend on x and is constant. For convenience, this sum is denoted as *2N*:(5)$$n_{1}\left(x,l\right)+n_{2}\left(x,l\right)=2N=const$$

The proton concentration and flux along the z-direction do not remain constant, since they are determined by diffusion and drift in the electric field which changes vs. the position of ions in the space of compartments [25]:(6)$$j=-D\frac{\partial n}{\partial z}+\frac{D}{h}n$$

As previously mentioned, the diffusion coefficient, *D*, takes on the values *D*_1_ and *D*_2_ on opposite sides of the membrane. We have also assumed that the membrane mainly passes protons and restricts the movement of other ions in the electrolyte. Therefore, the transport of protons determines the current and current density through the membrane and the entire cell.

An additional assumption is related to the diffusion coefficients of protons in both compartments, which are assumed to be equal since the electrolyte is a diluted solution. Therefore, *D*_1_ = *D*_2_. The parameter *h* in (6) represents a further usable special characteristic length that is defined as: (7)$$h=\frac{k_{b}T}{qE}$$
where *k_B_* is the Boltzmann constant, *T* is the absolute temperature of the electrolyte, and *q* is the ion charge.

The electric field, *E*, has different magnitudes *E*_1_ and *E*_2_ in the negative and positive compartments, respectively, while *E_m_* is the electric field inside the membrane. However, the electric field in all the cell’s elements (electrolyte bulk and membrane) is assumed to be constant. This assumption follows from the supposition of the infinite conductivity of the electrode’s material. The electric field magnitude will be estimated in the following Section 3.1. The sign of all electric fields is the same and can be either positive or negative. Therefore, the sign of the parameter h can be negative or positive, depending on the sign of the electric field.

The differentiation of the left side of Equation (6) with respect to z and substitution of this derivative into Equation (3) results in the following partial differential equation:(8)$$V\frac{\partial n}{\partial x}=-D\frac{\partial^{2}n}{\partial z^{2}}+\frac{D}{h}\frac{\partial n}{\partial z}$$

The dimensionless expression for Equation (8) takes the following form:(9)$$\frac{\partial^{2}\hat{n}}{\partial {\hat{z}}^{2}}-\frac{\partial \hat{n}}{\partial \hat{z}}+\frac{\partial \hat{n}}{\partial \hat{x}}=0$$
where:(10)$$\hat{n}=n/N$$
(11)$$\hat{z}=z/h$$
(12)$$\hat{x}=\frac{x}{\left(\frac{h^{2}V}{D}\right)}=\frac{x}{x_{0}}$$

It can be seen from (11) and (12) that h can be the characteristic scale of the concentration and flux changes along axis *z*, whereas $x_{0}=h^{2}V/D$ is chosen to be the characteristic scale for the proton concentration along axis x. It is worth reminding that only parameter x_0_ depends on the velocity *V*. The existing difference in proton concentration over the membrane’s upper and lower surfaces is diminished along with the x-coordinate because of the conservation law for protons and the hydrodynamic motion of electrolyte in the same direction. The increase in electrolyte motion prevents the diminishing of the proton concentration gradient along the x-axis.

The solution of Equations (8) and (9) will be found in the exponential form of plane waves:(13)$$\hat{n}=\left\{\begin{array}{c} C_{-}+A_{1-}e^{\lambda_{1}\left(\hat{z}-\hat{l}\right)-k_{1}\hat{x}}+A_{2-}e^{\lambda_{2}\left(\hat{z}-\hat{l}\right)-k_{2}\hat{x}},if\;0<z<l\\ C_{+}+A_{1+}e^{\lambda_{1}\left(\hat{z}-\hat{l}\right)-k_{1}\hat{x}}+A_{2+}e^{\lambda_{2}\left(\hat{z}-\hat{l}\right)-k_{2}\hat{x}},if\;l<z<2l \end{array}\right.$$

The coefficients *C*_−_ and *C_+_* in (13) represent the steady-state solution for the x→∞. Their values are equal to 1 as it follows from (5). The sign “−“ in Expression (13) corresponds to the negative electrolyte and “+” corresponds to the positive electrolyte compartment.

We can derive an expression for the proton flux in z-direction from (6) and (13):(14)$$\hat{j}=\left\{\begin{array}{c} 1+A_{1-}{\left(1-\lambda_{1}\right)e}^{\lambda_{1}\left(\hat{z}-\hat{l}\right)-k_{1}\hat{x}}+A_{2-}\left(1-\lambda_{2}\right)e^{\lambda_{2}\left(\hat{z}-\hat{l}\right)-k_{2}\hat{x}},if\;0<z<l\\ 1+A_{1+}{\left(1-\lambda_{1}\right)e}^{\lambda_{1}\left(\hat{z}-\hat{l}\right)-k_{1}\hat{x}}+A_{2+}\left(1-\lambda_{2}\right)e^{\lambda_{2}\left(\hat{z}-\hat{l}\right)-k_{2}\hat{x}},if\;l<z<2l \end{array}\right.$$

The unit of the proton flow density in (14) is *DN/h*.

The only solution when the proton concentration is diminishing with the electrolyte flow across the cell has physical meaning. Therefore, the only positive *k*_1_ and *k*_2_ should be considered for the solution of Equations (13) and (14). Moreover, the equality of $k_{1}=k_{2}=k$ has been taken into consideration due to the reason that the proton densities on electrodes and on both sides of the membrane should have similar dependence on the x-coordinate. This follows from the requirements to obtain a balance of forces applied to protons in the electrolyte solution.

The analysis of Equations (13) and (14) reveals that to obtain their final solution, there is the need to find seven unknown parameters: *A*_1−_, *A*_1+_, *A*_2−_, *A*_2+_, λ_1_, λ_2_, and *k*.

Two of the abovementioned parameters, λ_1_, λ_2_, can be related to *k* by the characteristic equation:(15)$$\lambda^{2}-\lambda -k=0$$
having the answer:(16)$$\lambda_{1}=\frac{1}{2}\left(1+\sqrt{1+4k}\right)\;\lambda_{2}=\frac{1}{2}\left(1-\sqrt{1+4k}\right)$$

Coefficient *k* will be determined further with regard to ensuring the physical meaning of the solution of Equation (13) for the proton concentration and the vicinity between theoretical and experimental data.

The five remaining parameters will be found from the three pairs of boundary conditions. The first one follows from the continuity of the proton flux though the membrane. According to (6), the proton flux through a membrane is equal to:(17)$$j=D_{m}\frac{n_{1}-n_{2}}{\delta_{m}}+D_{m}\frac{n_{m}}{h_{m}}$$
where *D_m_* is the coefficient of the proton’s diffusion, *n_m_* is an average proton density inside the membrane, *h_m_* is the characteristic length of their spatial distribution, and the parameter *δ_m_* represents membrane thickness. The characteristic length of *h_m_* depends on the electric field inside the membrane *E_m_* (see Equation (7)). At *z* = *l*, the proton fluxes (14) in both membrane sides should be equal to (17). The formulations for flux equality are represented in dimensionless form for convenience:(18)$$\left\{\begin{array}{c} -\frac{\partial {\hat{n}}_{1}}{\partial \hat{z}}+{\hat{n}}_{1}=r_{1}\left({\hat{n}}_{1}-{\hat{n}}_{2}\right)+r_{1}\frac{\delta_{m}}{h_{m}}{\hat{n}}_{m}\;\mathrm{if}\;z\to l,\;\;\;z<l\\ -\frac{\partial {\hat{n}}_{2}}{\partial \hat{z}}+{\hat{n}}_{2}=r_{2}\left({\hat{n}}_{1}-{\hat{n}}_{2}\right)+r_{2}\frac{\delta_{m}}{h_{m}}{\hat{n}}_{m}\;\mathrm{if}\;z\to l,\;z>l \end{array}\right.$$
where dimensionless variables ${\hat{n}}_{1},\;{\hat{n}}_{2},\;{\hat{n}}_{m},\;and\;\hat{z}$ are the results of the application of Expressions (10) and (11). Two combined relations (*r*_1_, *r*_2_) (18) are needed for the convenience of the following analysis:(19)$$r_{1}=\frac{D_{m}h_{1}}{D_{1}\delta_{m}}$$
(20)$$r_{2}=\frac{D_{m}h_{2}}{D_{2}\delta_{m}}$$
where *h*_1_ and *h*_2_ are the characteristic lengths of proton spatial distribution in the negative and positive cell’s compartments.

Expressions (18) represent the first pair of boundary conditions.

The substitution of dimensionless proton fluxes (14) into (18) and after the transformation with the appropriate consideration gives:(21)$$\left\{\begin{array}{c} r_{1}\frac{\delta_{m}}{h_{m}}{\hat{n}}_{m}=1\\ r_{2}\frac{\delta_{m}}{h_{m}}{\hat{n}}_{m}=1 \end{array}\right.$$

The equality of *h*_1_ = *h*_2_ follows from the (21) if *D*_1_ = *D*_2_.

The next set of boundary conditions is defined by the proton concentration at the electrode surfaces, specifically at *z * = 0 and *z* = 2*l*. These conditions are influenced by electrochemical reactions and control the electrical potential between electrodes [26]. However, in the current investigation, we set the boundary conditions and the electric potential between electrodes.

To ensure the independence of integration constants *A*_1_ and *A*_2_ over the x-coordinate in (13), the boundary conditions should have the following dimensionless form:(22)$$\left\{\begin{array}{c} {\hat{n}}_{1}(x,0)=1+\delta {\hat{n}}_{1}\exp\left(-k\hat{x}\right)\\ {\hat{n}}_{2}(x,2l)=1+\delta {\hat{n}}_{2}\exp\left(-k\hat{x}\right) \end{array}\right.$$

The constants of integration *A*_1_ and *A*_2_ should be independent over the z-coordinate at *x* = 0 as well. Therefore, the following boundary conditions should be determined:(23)$$\left\{\begin{array}{c} {\hat{n}}_{1}\left(0,z\right)=1+a_{1}\exp\left(\lambda_{1}\hat{z}\right)+a_{2}\exp\left(\lambda_{2}\hat{z}\right),\;\mathrm{if}\;0<z<l\\ {\hat{n}}_{2}\left(0,z\right)=1+a_{3}\exp\left(\lambda_{1}\hat{z}\right)+a_{4}\exp\left(\lambda_{2}\hat{z}\right),\;\mathrm{if}\;l<z<2l \end{array}\right.$$
where *a*_1_, *a*_2_, *a*_3_, and *a*_4_ are assumed to be arbitrary determined constants which will be chosen further. The boundary Conditions (22) and (23) should be consistent with each other at *x* = 0 and *z* = 0 or *z* = 2*l*. Therefore:(24)$$\left\{\begin{array}{c} \delta {\hat{n}}_{1}=a_{1}+a_{2}\\ \delta {\hat{n}}_{2}=a_{3}\exp\left(\lambda_{1}\frac{2l}{h_{2}}\right)+a_{4}\exp\left(\lambda_{2}\frac{2l}{h_{2}}\right) \end{array}\right.$$

Ultimately, the complete set of boundary Conditions (18), (22) and (23) along with their consistency (16) and (24) can determine the constants of integration in the Solution (13) of Equation (9).

## 3. Estimation of the Model Parameters

The final parameters of the cell’s electrical resistivity (conductivity) can be found from the solution of Equations (6), (8) and (9), determining proton flux and proton density inside the electrolyte. To achieve this goal, all initial parameters in boundary conditions (18), (22) and (23) and the model equations mentioned above, (6), (8) and (9), should be estimated.

### 3.1. The Assessment of the Electric Field Magnitude in the VRB

The assessment of electrical field magnitude was conducted with the following assumptions. First and foremost, it is important to consider the neutrality of the electrolyte solution, where a balance between positive and negative ions is maintained. This assumption holds true in theoretical studies, for instance in [23]. Assuming there are unknown surface charge densities *σ*_1_*, σ*_2_*, σ_m_*_1_*,* and *σ_m_*_2_ on the boundaries of the electrodes and membrane, the charge present on the membrane arises from two main factors. Firstly, it originates from the polarization of the membrane material, which is influenced by the electric field generated by the electrodes. Secondly, the precipitation of vanadium ions, which are unable to pass through the membrane, contributes to the charge distribution on the membrane.

It is important to note that we assume that the charges on the electrodes change slowly enough, allowing us to consider the electric field as stationary. This implies that the electric field does not depend on time and remains constant throughout the analysis. Dielectric constants inside the electrolyte and membrane are denoted as *ε*_1_, *ε*_2_, and *ε_m_*. We consider the permittivity of the vacuum and the dimensionless permittivity of the matter as a single factor, so *ε*_1_, *ε*_2_, and *ε_m_* are dimensional quantities. Since *L* and Y are much greater than *l*, we can assume that the charge density is weakly dependent on *x*, and the electric field is directed along the *z*-axis. By applying Gauss’s law, we obtain [27]:(25)$$\varepsilon_{1}E_{1}=\sigma_{1}$$
(26)$$\varepsilon_{2}E_{2}=\sigma_{1}+\sigma_{m1}+\sigma_{m2}$$
(27)$$\varepsilon_{m}E_{m}=\sigma_{1}+\sigma_{m1}$$
(28)$$0=\sigma_{1}+\sigma_{2}+\sigma_{m1}+\sigma_{m2}$$
where *E*_1_ and *E*_2_ are electric fields between the first and second electrode and the membrane, whereas *E*_3_ is the electric field inside the membrane.

Let the potential difference between the plates be equal to U, U > 0. Then, because $E=-\frac{\partial U}{\partial z}$ in a stationary one-dimensional electric field:(29)$$U=-E_{1}l-E_{2}l-E_{m}\delta_{m}$$

Note that the constancy of surface charge densities corresponds to a quasi-stationary state when heavy vanadium ions settle on the electrodes or on the membrane and do not actually participate in the charge transfer. In this case, the current through the battery is determined only by the transport of protons through the membrane.

To evaluate the current flow resulting from the movement of protons, it is necessary to analyze the electric field. This involves determining seven unknown variables. These variables include the magnitudes of three electric fields: two located between the electrode surfaces and the membrane and one within the membrane itself. Additionally, the variables encompass four charge densities on the electrode surfaces and both sides of the membrane.

However, it should be noted that only five equations, (25)–(29), have been presented thus far. The sixth equation can be derived from Equation (21), while the seventh equation is obtained from the analysis of the electrochemical reaction occurring on the electrodes. It is important to highlight that considering only one electrode reaction is sufficient for this analysis. For instance, during battery discharge, a reduction reaction takes place on the cathode where each ion ${\left(VO_{2}\right)}^{2+}$ is reduced to ${\left(VO\right)}^{+}$, resulting in the generation of two protons and the transfer of only one electron from the electrode to the solution. The increased concentration of ions near the electrode promotes their diffusion towards the opposing electrode. Simultaneously, protons can easily pass through the membrane, while vanadium ions tend to precipitate on the membrane, leading to the accumulation of surface electric charge.

Based on these observations and in accordance with electrochemical Equation (1), the following conclusions can be drawn regarding the positive charge on the cathode (*σ*_1_) and on the membrane surface opposite this electrode:(30)$${2\sigma_{1}=\sigma }_{m1}$$

This Equality (30) holds true in the steady-state operation of the cell, where for every electron transferred to the vanadium ions ${\left(VO_{2}\right)}^{2+}$ two protons are generated. During the electrochemical Reaction (1), it is important to note that each produced electron contributes to the generation of a charge with a magnitude equal to that of two protons.

The magnitudes of electric field strengths *E*_1_*, E*_2_*,* and *E_m_* can be estimated from (21) and (25)–(30) as:(31)$$E_{1}=E_{2}=-\frac{U}{2l+3\frac{\varepsilon }{\varepsilon_{m}}\delta_{m}}$$
(32)$$E_{m}=-\frac{U}{\delta_{m}}\frac{\varepsilon_{m}l+3\varepsilon \delta_{m}}{2\varepsilon_{m}l+3\varepsilon \delta_{m}}$$

Now, after the estimation of the strength of the electric field, the characteristic lengths for proton distribution in both compartments and within the membrane are outlined by Equations (7), (31) and (32):(33)$$h=-\frac{k_{B}T}{qU}\left(2l+3\frac{\varepsilon }{\varepsilon_{m}}\delta_{m}\right)$$
(34)$$h_{m}=-\frac{k_{B}T}{qU}\delta_{m}\frac{2\varepsilon_{m}l+3\varepsilon \delta_{m}}{\varepsilon_{m}l+3\varepsilon \delta_{m}}$$

### 3.2. Solution for the Proton Concentration and Flux in Cell Compartments

The solution to Equation (9) that determines the proton concentration is given by the Expression (13), which includes the constants of integration *A*_1−_, *A*_1+_, *A*_2−_, and *A*_2+._ These constants were derived from Equations (18) and (21)–(24) and have the following values:(35)$$A_{1-}=a_{1}\exp\left(\lambda_{1}\hat{l}\right),A_{1+}=a_{2}\exp\left(\lambda_{2}\hat{l}\right),A_{2-}=a_{3}\exp\left(\lambda_{1}\hat{l}\right),A_{2+}=a_{4}\exp\left(\lambda_{2}\hat{l}\right)$$

It is worth remembering that coefficients λ_1_ and λ_2_ can be obtained from the Expression (16) by the expression $\sqrt{1+4k\;}$ including variable *k*, which considering (16), (18) and (35) is equal to:(36)$$\sqrt{1+4k}=\frac{2r\left(a_{4}-a_{2}\right)\left(a_{1}-a_{3}\right)}{\left(a_{4}-a_{2}\right)a_{3}-\left(a_{1}-a_{3}\right)a_{4}}+\sqrt{1+{\left(\frac{2r\left(a_{4}-a_{2}\right)\left(a_{1}-a_{3}\right)}{\left(a_{4}-a_{2}\right)a_{3}-\left(a_{1}-a_{3}\right)a_{4}}\right)}^{2}}$$

To emphasize, it is important to note that the fulfillment Condition (36) guarantees the uninterrupted flow of protons across the surfaces of the membrane.

### 3.3. Formulation for the Entire Battery Current and Resistance

It is important to note that vanadium ions are unable to pass through the cation-exchange membrane. As a result, they do not contribute to the battery’s current and instead precipitate on the surface of the membrane. However, they do generate electric fields that influence the movement of protons. The external current of a battery, on the other hand, is produced solely by the movement of protons within the cells, which is facilitated by a proton-exchange membrane. This phenomenon can be attributed to several factors, which determine the entire battery current *J*:(37)$$J=q\int_{0}^{L_{Y}}dy\int_{0}^{L_{X}}j(x,z=l)dx=qL_{Y}\int_{0}^{L_{X}}j(x,z=l)dx$$
where *L_Y_* is the cell width in the y-axis (see Figure 2); we assumed that current density is constant alongside the y-coordinate due to the homogeneity consideration; *q* is the elementary charge, equal to $1.602\cdot 10^{-19}$*C*; *j*, the proton flux density in the cell, is determined either by the Equation (17) describing proton motion through the membrane or Equation (6) defining it in the compartment; and *L_X_* is the cell’s length along with the x-coordinate. Obviously, the flux density in (6) should be taken into consideration on the membrane boundaries, t.i. for *z* = *l* where both fluxes (6) and (17) are equal. Finally, battery current *J* can be represented by the electrolyte flow velocity in the x-direction if the expression of a proton flow in both compartments of a cell obtained from (14) is substituted into (37).

However, to solve the flow Equation (14) in the context of a vanadium redox flow battery, it is necessary to include appropriate boundary Conditions (18). The magnitude of the current can vary depending on factors such as the electrode design, flow rates, and electrochemical reactions taking place within the battery. The current generalized expression can be derived based on the assumptions used in the analysis of the vanadium redox flow battery, and it is represented by the following expression:(38)$$J=L_{Y}q\frac{VhN}{k}\left(\frac{kL_{X}D}{Vh^{2}}+\left(\left(1-\lambda_{1}\right)A_{1-}+\left(1-\lambda_{2}\right)A_{2-}\right)\left(1-\exp\left(-k\frac{L_{X}D}{Vh^{2}}\right)\right)\right)$$
where constants of integration *A_1−_* and *A_2−_* are determined by (35) with regards to parameters *k*, *λ*_1_, and *λ*_2_. It is worth mentioning that parameters *λ*_1_ and *λ*_2_ are defined by variable *k* through the Expression (16); D is the proton diffusion coefficient in the cell’s solution, equal in compartments, t.i. *D = D*_1_
*= D*_2_. Electrolyte flow velocity in (38) is designated as *VI*, and the summarized proton concentration of both membrane sides is denoted as *N*. The parameter *h* denotes the characteristic lengths for proton distribution in both compartments as in (7).

Summarizing the above, the battery resistance is equal due to the Ohm law:(39)$$R=\left|\frac{U}{J}\right|$$

The absolute value of the relation U to *J* is conditioned due to the possibility for the current to be either positive or negative depending on the charge–discharge mode.

### 3.4. Estimation of Model Parameters

To investigate the relationship between battery resistance and electrolyte flow speed, a numerical evaluation of the parameters incorporated into the model is necessary. However, it is important to note that certain parameters are assumed to be constants and were not altered during the development of the mathematical model. These constant parameters (*ε*, *ε_m_, D*, *D_m_*, and *N*) are considered fixed and do not change throughout the investigation. They are typically based on the technical literature and published data. The electrolyte and other relevant factors are deemed to remain unchanged.

Other constraints, such as *L_X_*, *L_Y_*, *l*, and *δ_m_*, were determined based on the dimensions of a vanadium battery. These parameters are specific to the design and geometry of the battery being studied. By incorporating these constraints into the mathematical model, it becomes possible to analyze the relationship between battery resistance and electrolyte flow speed within the context of a vanadium battery, considering the specific dimensions and characteristics of the system.

Parameters such as U (potential difference between electrodes) and *T* (absolute temperature of the electrolyte) are determined based on the specific experimental conditions. These parameters are directly measured or controlled during the experimental setup.

The potential difference, U, is typically applied across the electrodes to drive the electrochemical reactions within the battery. It plays a crucial role in determining the performance and behavior of the battery system. The value of U is determined based on the experimental design and requirements.

The absolute temperature, *T*, of the electrolyte is an important parameter that affects various electrochemical processes within the battery. It influences the conductivity of the electrolyte, the kinetics of the reactions, and the overall performance of the battery. The value of *T* is measured or controlled using appropriate temperature monitoring equipment.

Additionally, there are other parameters (*a*_1_, *a*_2_, *a*_3_, *a*_4_) that are initially unknown but play a crucial role in determining the accuracy of the model with respect to real experimental results [28]. These experimental data are represented in the Appendix A. These parameters are typically determined using a specialized mathematical fitting procedure.

The mathematical fitting procedure involves comparing the predictions of the model with the actual experimental data and adjusting the values of these unknown parameters to minimize the difference between them. This process helps calibrate the model to accurately represent the behavior of the specific battery system under investigation.

The specialized mathematical fitting procedure can be based on optimization techniques such as least squares regression, which was chosen in the work. By iteratively adjusting the values of the unknown parameters, the fitting procedure finds the optimal set of parameter values that best align the model predictions with the experimental results.

The determination of these unknown parameters (*a*_1_, *a*_2_, *a*_3_, *a*_4_) using a mathematical fitting procedure is crucial for ensuring the model’s accuracy and reliability in representing the real experimental behavior of the battery system.

All the aforementioned variables are listed in Table 1 and are subsequently utilized to estimate the battery resistance, as well as the spatial distribution of proton concentration and current density. The selection of parameters in Table 1 is grounded in our prior empirical investigation [28], supplemented by the contemporary literature that presents representative values for those parameters that were not explicitly measured in the aforementioned study.

The step-to-step algorithm of a fitting process for the estimation (*a*_1_, *a*_2_, *a*_3_, *a*_4_) is described below:*a*_1_ = 0. It follows from three conditions. The first is that the coefficient *r* should be assigned a negative value as it is determined by the sign of the electric field, which is assumed to be negative in our investigation. The second condition is that the expression for the proton concentration (13) must always yield a positive value. The third condition is that the magnitude of the membrane current density should be kept below the allowable limit (~300 mA/cm^2^), as reported in [28,33,34].The sequential choosing of *a_2_* in the range of more than 0 and less than 1, t.i. *a*_2_: 0 < *a*_2_ < 1. This is the consequence of the fact that *δn*_1_ should be less than *N* in consistency Condition (24).The sequential choosing of *a*_4_ among the range from (0.1∙*a*_2_) to *a*_2_. The reason for this is that the coefficient *k* should be assigned a positive value, as it aligns with the physical consideration of the decreasing proton concentration along the *x*-axis.Calculation of *k* using the Equality (36).The coefficients must adhere to the equality which is a consequence of (5) and (13):
(40)$$\left(a_{1}+a_{3}\right)\exp\left(\sqrt{1+4k}\frac{l}{h}\right)+a_{2}+a_{4}=0$$Calculation of *a*_3_ using Expression (40).Calculation of *λ*_1_ and *λ*_2_ based on (16).Calculation of the constants of integration *A*_1−_, *A*_1+_, *A*_2−_, and *A*_2+_ with (35).Calculation of total current through the membrane using (38).The array of resistances of the vanadium cell for different electrolyte speeds is estimated by (39), and then the summarized error between modeled and real magnitudes is assessed.Compute model predictions: Using the current parameter values, compute the model predictions for the given experimental conditions. This involves solving the mathematical model that relates the battery resistance and electrolyte flow speed to the unknown parameters and other known variables.Evaluate the least mean square (LMS) for chosen *a*_2_ and *a*_4_ values providing a theoretical resistance curve with those obtained by the measurements. The LMS quantifies the mismatch between the model and the experimental data.Choosing the next combination of *a*_2_ and *a*_4_ values. Then, use the abovementioned algorithm from steps (*a*) to (*l*) to find the best combination of *a*_2_ and *a*_4_ values providing the minimum of LMS. The algorithm adjusts the values in a direction that minimizes the objective function, gradually improving the fit between the model and the experimental data.The combination of all variable parameters leading to the minimum of the summarized error (LMS) is recognized as the optimal for the math model.

## 4. Results

The main research outcome of this study focuses on investigating the influence of electrolyte speed increase on the reduction in the battery’s internal electrical resistance. This relationship is examined using a theoretical model, and the results are compared with experimental data, as depicted in Figure 4. The comparison between the theoretical model and the experimental results demonstrates the agreement between the two. The theoretical curve, which represents the predicted behavior based on the proposed theoretical explanation, closely aligns with the actual resistance of the cell obtained from the experimental measurements. To quantify the accuracy of the theoretical model, the mean squared error (MSE) is calculated by the following expression:(41)$$MSE=\frac{\sqrt{\frac{1}{M}\sum_{i=1}^{M}{\left(R_{Th\_i}-R_{\exp\_i}\right)}^{2}}}{{\left(R_{Th}\right)}_{average}}$$
where *M* is the total number of experimental points, *i* denotes the indexes, and *R_Th_* and *R_exp_* are the theoretical and experimental resistances.

The MSE measures the average squared difference between the theoretical curve and the actual resistance values. The comparison showed a good accuracy of the developed theoretical model, which can be estimated by the MSE value that is less than 3.3%.

The small value of the MSE indicates a strong level of agreement between the theoretical predictions and the experimental results. It suggests that the proposed theoretical explanation provides a highly accurate representation of the influence of electrolyte speed increase on the reduction in the battery’s internal electrical resistance.

These findings offer valuable insights into the relationship between electrolyte flow speed and battery resistance. The high accuracy of the theoretical model further strengthens the validity of the proposed explanation, enhancing confidence in its applicability to real-world scenarios.

Based on the previously validated theoretical model (as demonstrated in the comparison presented in Figure 4), an analysis of the battery resistance was conducted considering dissimilar cell lengths. The outcomes of this analysis are illustrated in Figure 5.

Figure 5 showcases the relationship between the battery resistance and the varying electrolyte velocity. The theoretical model, based on the proposed explanation, predicts how changes in the velocity affect the internal electrical resistance of the battery.

By examining Figure 5, it becomes evident how different electrolyte speeds impact the resistance of the battery. The results highlight the trend or pattern between electrolyte flow and resistance, providing valuable insights into the behavior of the system.

To facilitate the analysis, the flow speed of the electrolyte was divided by a characteristic speed, which was determined by:(42)$$V^{*}=\frac{kLD}{h^{2}}$$

For the specific vanadium battery under consideration, the characteristic speed is approximately 11 cm/s. The analysis results indicate that the trend of decreasing battery resistance becomes less significant and levels off as the speed increases for each battery design. It is evident that surpassing the characteristic speed ($V^{*}$) is impractical, as it offers limited benefits in terms of further reducing the battery resistance. This observation holds importance for optimizing the electrolyte flow in the battery’s design.

Another factor that contributes to a decrease in battery resistance is the length of the cell in the direction of the electrolyte flow. The provided analysis shows the positive influence of the cell’s length (and the membrane external surface) in the direction of the electrolyte flow on the battery’s electrical conductivity. However, the tendency of conductivity increase flattens since during the electrolyte flow the concentration of protons becomes lower (see Figure 6). Graphs of the resistance are represented by the natural logarithm in the dimensionless form equal to the *R/R_max_*, where *R_max_* is assumed to be 0.014 Ω.

Additional important characteristics influencing the battery’s functionality and electrical resistance are represented by the spatial distribution of protons and current densities, which are shown in Figure 7 and Figure 8.

The analysis shows that proton concentration achieves its maximum near the positive electrode. This study demonstrates that the similar tendency of the current density coincided with the proton concentration change.

The graph in Figure 8 demonstrates the distribution of the absolute value of vector of the dimensionless proton current density which is directed in the *z*-axis. The proton movement in the *x*-axis takes place with the electrolyte flow but it is less important since it does not participate in electrochemical reactions. The unitless magnitudes of these vectors obtained by the division of their values on the base current density equal to qDN/h are the function of the x- and z-coordinates. The maximum current density is typically observed near the cathode in a redox flow battery. The distribution of current density exhibits a surge on both sides of the ion-exchange membrane, while it changes continuously within the membrane’s interior as it follows from the boundary conditions (18).

## 5. Conclusions

The objective of this investigation was to examine how the velocity of the electrolyte affects the resistance of a vanadium redox flow battery (VRB). To accomplish this goal, a mathematical model was developed, which consisted of a system of partial differential and algebraic equations describing the two-dimensional behavior of the vanadium cell. The model analyzed the steady-state processes occurring within the cell, considering proton flow through diffusion, drift, and mass transportation, as well as the movement of protons through the ion-exchange membrane. The model also considered the influence of charge densities on the surfaces of the electrodes and membrane, as well as the impact of electrochemical reactions on battery resistance. To solve the equations, appropriate boundary conditions were defined for the electrodes, membrane surfaces, and the entry of the electrolyte into the cells. The variables and coefficients of the model, as well as the boundary conditions, were determined by the special software procedure ensuring the best closeness between the theoretical description and experimental data. The value of the MSE is less than 3.3%.

By incorporating these variables and coefficients into the mathematical description, the theoretical dependence of battery resistance vs. electrolyte velocity was modelled, providing insights into the precise behavior of the battery and its resistance. In addition, the characteristic velocity of the electrolyte movement is estimated at a significant excess of which the resistance ceases to depend on the velocity. In our case, the velocity is equal to 11 cm/s.

The investigation showed the non-linear dependence of battery resistance on electrolyte flow velocity and cell dimensions.

This investigation aimed to elucidate the relationship between electrolyte velocity and VRB resistance, contributing to the understanding and optimization of vanadium redox flow batteries for practical applications.

The represented investigative work conducted on vanadium redox batteries to analyze the impact of electrolyte flow velocity on resistivity has broader implications beyond just VRBs. The findings and methodology can be successfully applied to other types of flow batteries as well. This suggests that the insights gained from this study can contribute to the development and optimization of various flow battery technologies, enhancing their performance and efficiency. Further research in this area could explore the applicability of these findings across different flow battery systems and aid in advancing the overall understanding of flow battery operation and design.

## Figures and Tables

**Figure 1 materials-16-04845-f001:**
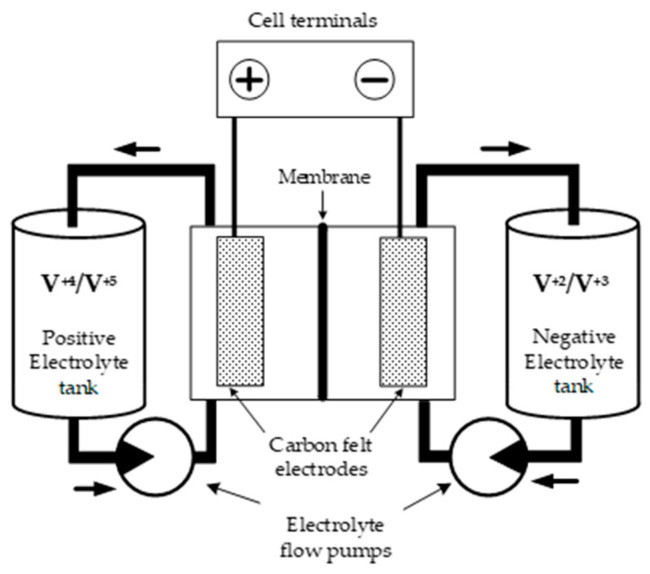
The principal design of a vanadium cell.

**Figure 2 materials-16-04845-f002:**
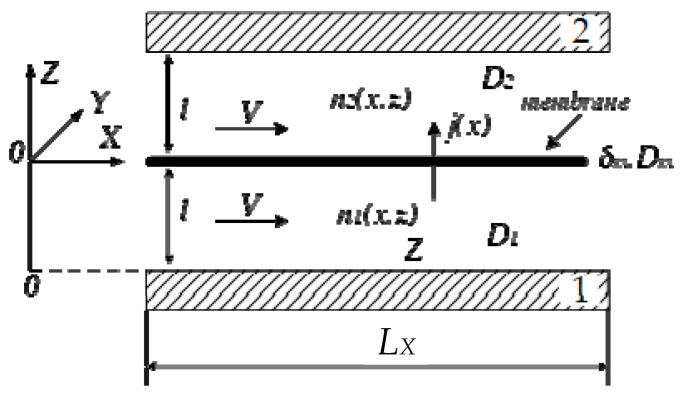
The principal sectional diagram of a cell with electrolyte flow parallel to the membrane surface: 1,2 electrodes of vanadium redox cell.

**Figure 3 materials-16-04845-f003:**
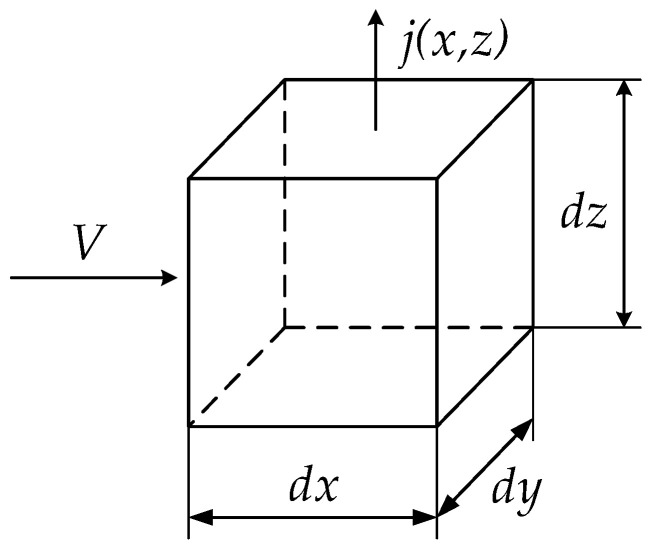
The arbitrary chosen parallelepiped (*dx*, *dy*, and *dz*) of the elemental electrolyte volume whose faces are parallel to the plane of the electrodes and the membrane and axes x, y, and z. The velocity of the electrolyte is denoted as V, and proton flux is denoted as *j*(*x*,*z*).

**Figure 4 materials-16-04845-f004:**
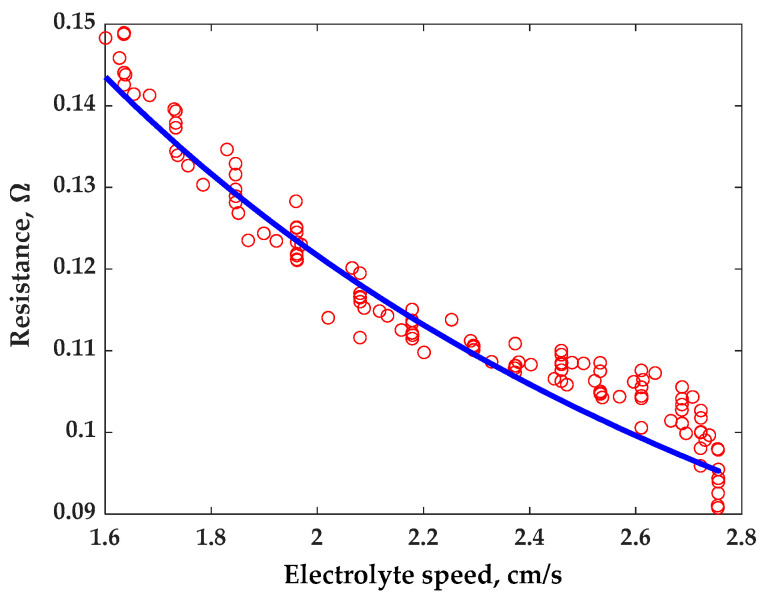
Battery cell theoretical and experimental internal resistance vs. electrolyte speed. The blue line designates the theoretical curve, whereas the experimental results are represented by the red circles.

**Figure 5 materials-16-04845-f005:**
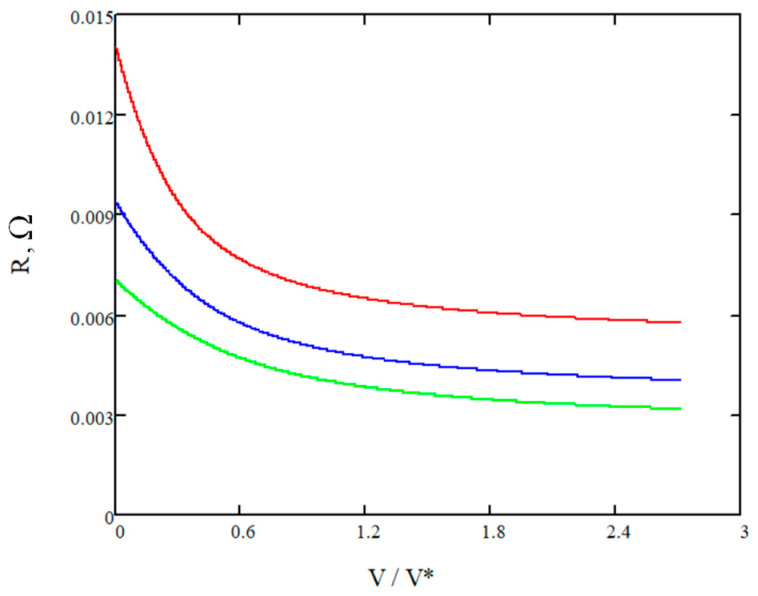
The dependence of the resistance vs. vanadium cell’s length: red—80 cm, blue—120 cm, green—160 cm.

**Figure 6 materials-16-04845-f006:**
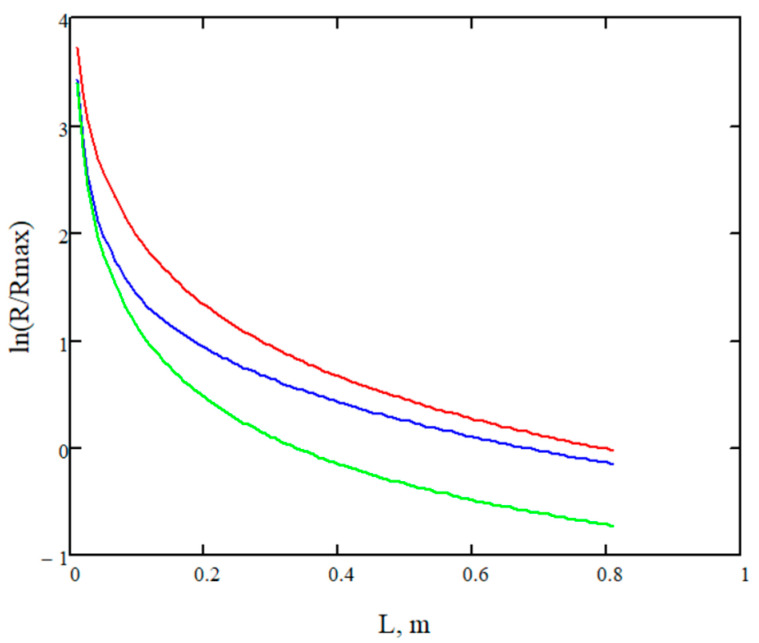
The natural logarithm of the relative battery resistance vs. cell’s length for different electrolyte speeds: the red—0.1 cm/s, blue—1 cm/s, green—10 cm/s.

**Figure 7 materials-16-04845-f007:**
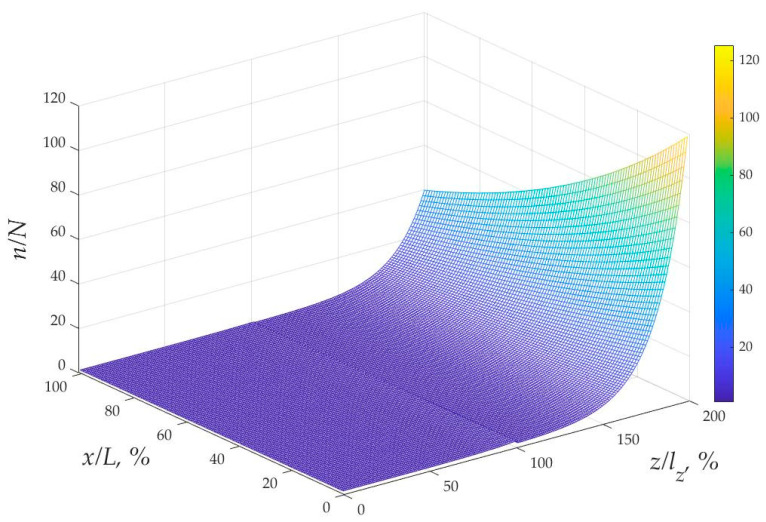
The normalized 2-D concentration of protons (*n*/*N*) vs. *x*/*L_x_*, *z*/*l* coordinates expressed in dimensionless form obtained for the electrolyte speed 10 cm/s.

**Figure 8 materials-16-04845-f008:**
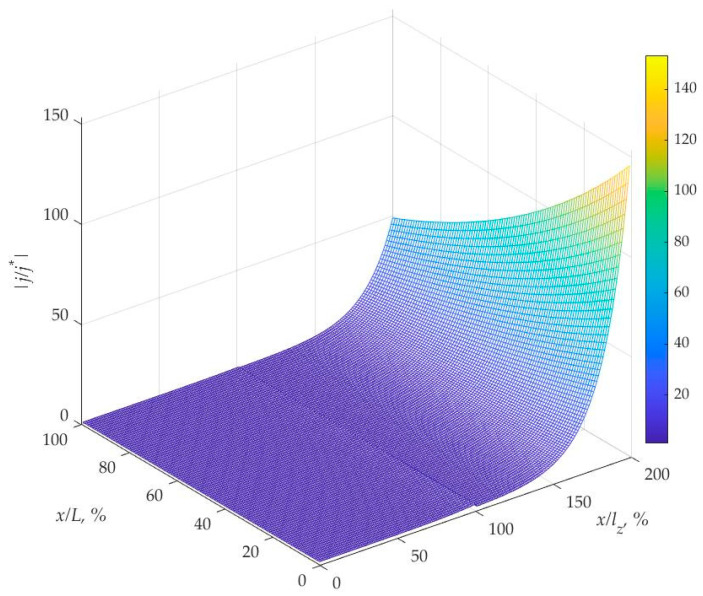
The current density vs. *x*/*L_x_*, *z*/*l* coordinates expressed in dimensionless form obtained for the electrolyte speed 10 cm/s. j/j* is the dimensionless current density, j*=qDN/h is the base current density.

**Table 1 materials-16-04845-t001:** List of the primary parameters of the model.

Parameter	Value	Units	References
*ε*	81∙*ε*_0_	* _-_ *	[29]
*ε_m_*	78∙*ε*_0_	*-*	[29]
*L_X_*	800	mm	[28]
*l*	3.5	mm	[28]
*L_Y_*	600	mm	[28]
*δ_m_*	0.03	mm	[13,28]
*T*	318	K	[28]
*D*	9∙10^−9^	m^2^/s	[30,31,32]
*D_m_*	5.1 ∙ 10^−10^	m^2^/s	[20]
U	1.455	V	[28]
*V*	0–0.3	m/s	[28]
*N*	2∙10^25^	m^−3^	[18,31]
*a* _1_	0	-	
*a* _2_	4∙10^−3^	-	
*a* _3_	−1.722 ∙ 10^−19^	-	
*a* _4_	1.000 ∙ 10^−3^	-	

## Data Availability

Data is contained within the article and the Appendix A.

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
