# Peer review of "Investigating the Impact of Electrolyte Flow Velocity on the Resistivity of Vanadium Redox Batteries: A Theoretical Analysis and Experimental Data Comparison"

_materials, 2023, doi:10.3390/ma16134845_

Round 1
Reviewer 1 Report
This article presents novel model for more thorough knowledge on flow batteries efficiency. However, the following issues should be addressed before publishing:
1. Adjust the references and check spelling throughout the Manuscript in accordance with Materials template.
2. In lines 71-75, you reference number of researches, add more details about their results.
3. Start new paragraph from line 112.
4. Lines 121-123 are unnecessary.
5. In Table 1. ‘List of the primary parameters of the model’ you have named references from which experimental data were used. Why did you choose these studies?
6. Could this model be applied to other flow batteries? Could you emphasize the potential of your research for future investigations?
Author Response
Dear Reviewer,
Thank you for your fruitful work dedicated to improving our manuscript.
Your remarks with our answers are represented in the attached file.
Thank you once again,
Corresponding authors

Reviewer 2 Report
1. Title:
The current title is kind of vague since it the “conclusion” of this work. However, a good title should let the reader catch what specific work it is in which filed, and it is not showing up here. Please rewrite an appropriate title to explain to the reader what kind of modeling work you did in which field.
2. Abstract
Since it is a modeling work, please select appropriate language to describe the conclusion in the introduction part. In row 13 and 14, the author claims the electrolyte velocity could be a key factor to improve the cell efficiency. However, this sentence is confusing the reader since it is not clear here whether this is only modeling result or modeling with the experiment.
3. Introduction
I greatly appreciate the author's detailed introduction to the work function of the vanadium cell. However, in order to enhance the clarity and cohesiveness of the article, I suggest revising paragraphs from row 66 to row 119. Specifically, focusing on the latter half of the paragraph (row 100 to row 119) appears to present the information more logically. To achieve a clearer discussion, it would be beneficial to incorporate relevant studies and expound upon the motivation behind our work. Also please citing pertinent research and explaining the purpose of yourstudy more concisely.
4. The modeling of VR cell
4.1 Figure 2. Please correlate this sectional diagram with figure 1 and explain how the battery works here so reader can catch up easily why this diagram been define such way.
4.2 In the section 2.1 assumption part, author mainly focus on define the internal electrical environment by define proton distribution and internal electrical field by using math. However, if author think this is the appropriate way, please provide enough assumption/evidence from chemistry/fluid mechanics side to provide these factors could be fixed when you are providing the model in such way.
4.3 In the section 3.1, row 357-row 362 are duplicated from row 367-372. Please also explain how the reaction happened during the charge process.
4.4 row 379, please put this equation in next row.
4.5 Kindly suggest the author can provide a real case (example) in the estimation of model parameters section regarding how different parameters been defined as supplementary material—since the experiment data been provided in the next section.
This include Dimension based parameters; Experimental condition based parameters; mathematical fitting parameters
4.6 please double check the row 484, what is the right factor of a2.
4.7 Kindly suggest to provide a real case as supplementary material for step-to-step fitting process, particular the step m.
4.8 Kindly suggest to provide how the 3.3% MSE calculated.
5.Results
I think the Author spends lots of text to explain how this model been build and how the parameters involved. However this “discussion” section is relative weak. The detail parameters of such fitting are mot provided and the experiment data in figure 4 are also not provided. Kindly suggest author can provide more information here. There is a clear “gap” between the conclusion and the modeling need to be fulfilled.
Also kindly suggest the author can provide comparison of this PDE based, electrical filed focused model with models (such as chemistry or flow mechanics, FEA driving modeling)from other works in specific way. Reader will have a more comprehensive understanding of which method should be selected based on purpose.
6.Discusion
The most accurate name of this section should be “conclusion”.
Author Response

(The authors gave the same response as above.)

Round 2
Reviewer 2 Report
The authors addressed the questions mentioned in the first round review. Now the manuscript is sufficiently improved. The reviewer would like to recommend the publication of this manuscript in the present form